# Early Nurse Management Experiences from Finnish COVID-19 Hubs: An In-Action Review

**DOI:** 10.3390/ijerph19084885

**Published:** 2022-04-17

**Authors:** Mari S. Nevala, Satu Vuorela

**Affiliations:** Laurea University of Applied Sciences, 01300 Vantaa, Finland; satu.vuorela@laurea.fi

**Keywords:** nurse management, crisis management, primary healthcare, COVID-19 ambulatory care, in-action review

## Abstract

Primary healthcare (PHC) clinics are the point of access for many COVID-19 patients; however, the focus of crisis response work has been in securing hospital capacities. The purpose of this study was to describe the early nurse management experiences from PHC clinics within Greater Helsinki dedicated to caring for all ambulatory patients with possible COVID-19 symptoms. The study objectives were to make PHC crisis response contributions known and to provide an in-action review (IAR) of crisis response efforts. Nurse managers from the four COVID-19 hubs in Greater Helsinki were interviewed using thematic pair interviews. The data were analyzed inductively using thematic analysis, by which four main themes emerged: (1) capacity development led to a state of flux, (2) infection prevention control (IPC) was critical, (3) management of staff was essential in facilitating crisis response, and (4) respondents’ personal experiences. The state of flux stressed the provision of PHC services, but quick developments in telemedicine eased that burden. Conversation surrounding IPC was extensive, though discrepancies suggest that global efforts to standardize IPC practices must begin locally. Leadership was adjusted to accommodate for the crisis, especially regarding the motivation of staff. A vision to aspire toward in crisis recovery is needed.

## 1. Introduction

In December 2019, an outbreak of pneumonia with an unknown etiology occurred in the city of Wuhan, China. Scientists later identified the cause of the viral disease now known as the coronavirus disease 2019 (COVID-19) to be a novel strain of coronavirus (SARS-CoV-2) [1]. On 11 March 2020, the World Health Organization (WHO) declared COVID-19 a pandemic. At that time, Finland had confirmed a total of 40 COVID-19 cases and zero deaths associated with the disease [2]. Though Finland had a relatively small number of cases early in the pandemic, examples of overwhelmed healthcare systems in China and Italy provided data to model pandemic situations and their effects on healthcare capacities [3]. The Finnish Institute for Health and Welfare (THL) had predicted that the peak of the first pandemic wave could potentially result in over 900 simultaneously hospitalized COVID-19 patients in Finland, of which 280 would need intensive care unit (ICU) level care [4]. Before the crisis, Finland had a total of 259 ICU beds, of which 70 were in the capital region [5]. According to the models, COVID-19 could have become a substantial threat to healthcare capacity in Finland [3].

The government of Finland declared emergency conditions on the 16 March 2020, based on the fifth exceptional situation defined by Finnish law, in which a dangerous and contagious pathogen is widespread [6,7]. In order to slow the spread of the virus within the population, the government implemented nationwide restrictions. The Finnish government also defined crisis standards of care, by which healthcare organizations postponed non-essential services and prioritized urgent care. The Ministry of Social Affairs and Health (STM) published guidelines for surge capacity development in emergency and ICU services, requiring 1.5–2 times the normal ICU capacity. During the peak of the first pandemic wave, there were consistently over 40 (<50) COVID-19 patients in the ICUs within the capital region, meaning that the increase in ICU capacity was more than sufficient [5].

Countries that have been successful in slowing the progression of the COVID-19 pandemic have implemented healthcare systems that include diagnosing and treating mild cases [8]. However, the focus of crisis response work has primarily been on hospital capacities. The STM had instructed municipalities to dedicate an ambulatory access point, referred to as COVID-19 hubs in this study, for suspected and confirmed cases of COVID-19 [9,10]. The three cities within Greater Helsinki each dedicated entire primary healthcare (PHC) clinics as their COVID-19 hubs [11].

Primary health care is the frontline defense for COVID-19 crisis response work at the community level [12,13,14]. The WHO defines PHC to be health care that accounts for the whole of society, considering both the proximity and equitable distribution of healthcare services [15]. The Finnish Health Care Act divides Finland’s healthcare system into two distinct areas, provincial and municipal, or specialized and PHC services [16]. Before the PHC development in 1972, 90% of healthcare costs in Finland were due to specialized services. Now, even with just 0.7 physicians in PHC per 1000 inhabitants, specialized health care accounts for just 5% of total healthcare costs. Nurses’ roles in PHC are important, as nurses held a total of 28% of urgent care-related PHC appointments [17]. These numbers highlight the importance of PHC and PHC nurses in the Finnish healthcare system and, ultimately, the significance of their contribution to the national COVID-19 crisis response as well.

### Literature Review

The WHO has classified all three coronavirus diseases as priority diseases for research and development because of their risk to public health, which makes infection prevention and control (IPC) critical [18]. The WHO defines IPC to be a practical discipline that aims at preventing infections in healthcare settings, and for which implementation depends upon the transmission mode of a given pathogen [19]. COVID-19 is defined as being transmissible through droplets, meaning that it spreads through infected liquid respiratory particles [20]. Transmission risk management is the mitigation of transmission risk to as low as reasonably possible without unintentional repercussions, and a hierarchy of controls identifies relevant interventions, which are, in the descending order: elimination, substitution, engineering, administrative, and personal protective equipment (PPE) controls [21,22].

Elimination, within the hierarchy of controls, is the most effective and means removing the risk altogether—for instance, through physical isolation of symptomatic patients. Substitution refers to replacing the risk with another risk [22,23,24]. Engineering controls remove or reduce the hazard at its source, such as by maintaining physical distance through seating arrangements, improving ventilation and air filtration, and using physical barriers such as plexiglass partitions [20,22,25]. Administrative controls change working patterns to reduce or eliminate transmission risk, and leaders within health care are responsible for creating, communicating, and implementing clear IPC policies, such as policies for managing ill staff utilizing syndromic surveillance [19,22,25,26,27,28]. Personal protective equipment is the least effective at protecting healthcare workers (HCW) from risk and there has been much discussion about the level of PPE needed to mitigate nosocomial COVID-19 transmissions [21,23,29]. According to the THL, HCWs are to comply with universal, contact, and droplet precautions when caring for suspected and confirmed cases of COVID-19 [30]. The National Emergency Supply Agency was responsible for securing PPE when shortages threatened to put HCWs at risk [31].

Frontline HCWS are susceptible to both unprecedented physical and psychological stress related to the crisis, and many nursing platforms are worried about the wellbeing of nurses [32,33,34,35,36,37]. The COVID-19 healthcare crisis has underscored the need for effective leadership within health care, and understanding what motivates and empowers HCWs to work during public health emergencies, despite the risks involved, is an important aspect of preparedness [32,38]. One review of nurses’ experiences in working during acute respiratory infection (ARI) epidemics found that nurses have a personal sense of professional obligation to work despite the risks, but are supported through professional camaraderie, adequate PPE and staffing, clear and timely communication of information, organizational preparedness, and effective leadership [39]. Nurse leaders identified that the most challenging aspect of COVID-19 leadership is responding to the increase in emotional health needs of nurses, stating that leadership in nursing may face challenges, such as distrust, in the future due to COVID-19 experiences [40]. Servant leadership, which is a leadership style in which leaders are defined by their desire to serve others and by their intent to empower their followers, can be argued to be an effective leadership style in healthcare crises, as frontline HCWs are served by their leaders in order to continue to provide quality patient care [41,42]. It is also an effective leadership method in building trust between leaders and followers [41].

Crisis management is the development and implementation of an organization’s capability to respond to and recover from crises, as insufficient crisis recovery efforts can lead to a new crisis [43]. The WHO identifies PHC to be a gatekeeper in the COVID-19 healthcare crisis by reducing the need for hospital services via the identification, triage, and diagnosis of COVID-19 patients and by addressing the fear of COVID-19 within the community [12]. Primary health care has been immeasurably burdened by the crisis, due to factors such as the prioritization of COVID-19-related care [44,45]. The PHC contribution to the COVID-19 crisis response is seen in the literature through after-action reviews, which show both the resilience and weaknesses of our healthcare systems [10,14,44,46,47]. Frontline PHC nursing experiences are less represented in the literature, though one study found issues concerning employment insecurity and inadequate PPE to be elements of the PHC nursing COVID-19 experience [48]. There are very few, if any, publications on the experiences of nurses and/or nurse managers working within COVID-19 hubs.

In order to evaluate and improve public health response, both after-action reviews (AAR) and in-action reviews (IAR) of crisis response efforts are essential [49,50,51]. The purpose of this study was to describe the early nurse management experiences from PHC clinics within Greater Helsinki dedicated to caring for all ambulatory patients with possible COVID-19 symptoms. The research question was: “What were the early experiences of nurse managers of PHC clinics dedicated to caring for patients with possible COVID-19 symptoms?” The objectives were to make PHC crisis response contributions known and to provide an IAR of crisis response efforts. Nurse managers from the four COVID-19 hubs within Greater Helsinki were interviewed using thematic pair interviews. Data were analyzed inductively using thematic analysis. This study was conducted as the first author’s master’s thesis in Global Health and Crisis Management [52].

## 2. Materials and Methods

Both AAR and IAR guidelines were utilized for the purposes of this study [49,50,51]. The data were collected through thematic pair interviews of key informants using telecommunication. The study was designed to use minimal staff resources and to reduce the role of the interviewer to lessen the impact of possible bias. Thematic interviewing is defined to be conversation encompassing predefined themes, for which the structure of the interview framework is essential [53]. In generating the themes for this interview, a semi-structured literature search was conducted on the topic with the help of information retrieval guidance services provided by the university library. Themes were extracted by categorizing information into possible background information and potential interview themes. Potential interview themed notes were coded, and the codes were then categorized into five themes: capacity development, management of staff, infection prevention, emerging issues, and overall experiences. Possible questions were then derived from the codes.

A clear purposive strategy for respondent recruitment was applied using the defined target group from the purpose of the study [53]. The term “early” is defined by the time between the activation of the Emergency Powers Act and the last conducted interview (16 March 2020–2 February 2021). The respondents were all working as nurse managers of the four COVID-19 hubs in Greater Helsinki during this time. Greater Helsinki is defined by the three separate municipalities of Espoo, Helsinki, and Vantaa. Espoo and Vantaa each had one COVID-19 hub, while Helsinki had two. The four respondents were identified as the most relevant sources of information because including respondents from other parts of Finland would have diluted the context of the study. The first author contacted the respondents directly using the names and email addresses listed on municipal websites [53]. All communication with the respondents, including the interviews, was conducted in Finnish. All three cities gave their approval for the study and all four targeted nurse managers participated in the study. To better triangulate the data, the two managers from Helsinki were each paired with a manager from either Espoo or Vantaa.

In line with general data protection regulations (GDPR), each respondent gave their written and informed consent to the collection, usage, and storage of the data produced by the interviews. Only the first author had access to the original data from the interviews. After the thesis report was published in December 2021, the original data from the interviews (recordings, transcriptions, and original paper codes) were destroyed due to the high risk of being able to identify respondents and their contributions. The original mind maps are retained by the first author. Respondents were made aware of the extent of their contribution through member checking and participation was voluntary throughout the duration of the study [53,54]. This study was exempt from an ethical statement from the Human Sciences Ethics Committee of the Helsinki Region of Universities of Applied Sciences; however, the first author retains the study permits obtained from the municipalities and the informed consent letters signed by the respondents [55].

### 2.1. Data Analysis

The data were analyzed and coded inductively using thematic analysis. Thematic analysis is a qualitative research analysis method that can be used to identify, analyze, organize, describe, and report themes within data. The thematic analysis process can be described in six phases [56].

#### 2.1.1. Phase 1: Familiarize with the Data

This phase requires an immersion into the data, which is achieved through active rereading of the data [56]. After the data were collected, the recorded interviews were listened to by the first author multiple times before, during, and after transcription. The transcription process itself engendered active familiarization with the data, as the first author transcribed both interviews herself. Throughout the analysis process, the first author would return to the full transcriptions and/or interview recordings as needed. This was done when the context of authentic separated data was in question.

#### 2.1.2. Phase 2: Form Codes from the Data

Coding happens after an in-depth familiarization of the data and is a reflective process that simplifies data characteristics [56]. This second phase was completed by hand [53]. Two copies of each interview transcription were printed at the university library. Initial codes were identified and documented on one copy by identifying sections of text belonging to a certain code. These sections were then cut out of the second copy of the transcriptions and labeled for temporal placement in the interview. The cut-out text sections were attached by paper clips to index cards labeled with the identifying codes. The index cards were placed in front of the first author as she continued to code, making it easier to add text sections to codes because the data included four similar accounts of experiences. In this way, the coding was not entirely inductive but contained deductive properties as well. Initially, the unit of analysis was keywords and phrases, but due to fragmentation, the data were later recoded using meaning units.

#### 2.1.3. Phase 3: Form Themes from the Codes

Themes combined single components of data and were generated inductively by arranging and rearranging the codes into groups of similar codes [56]. The author continued to refamiliarize herself with the authentic data, and mind maps were utilized in linking codes together into themes. The mind maps produced hierarchies of codes, categories, themes, and main themes.

#### 2.1.4. Phase 4: Review the Themes

The coded data was then compared to the themes to make sure there was a coherent connection between them [56]. The author refamiliarized herself with the authentic data in the form of the full transcriptions and interviews, while asking the data “What belongs under this theme?” Some themes expanded or collapsed, though always in reference to the authentic data. This phase was conducted until successful member checking and theoretical saturation were reached.

#### 2.1.5. Phase 5: Name and Define the Themes

The fifth phase involved defining and naming themes. This was completed by determining what parts of the data were captured within each theme [56]. Once all relevant sections of the data were included within themes, the analysis was complete. The four main themes that emerged from the analysis are state of flux, infection prevention, management of staff, and personal experiences.

#### 2.1.6. Phase 6: Writing the Results

The data analysis was conducted in the original language of the content. Results were translated by the first author during the writing of the report. The translation from Finnish to English was delayed for as long as possible due to its ontological significance [57]. The results were written utilizing the final mind maps of the four main themes. Each direct quote has been used with the permission of the corresponding respondent.

## 3. Results

The purpose of this study was to describe the early nurse management experiences from PHC clinics within Greater Helsinki dedicated to caring for all ambulatory patients with possible COVID-19 symptoms. The research question was: “What were the early experiences of nurse managers of PHC clinics dedicated to caring for patients with possible COVID-19 symptoms?”. Two thematic pair interviews were conducted by the first author on 10 November 2020 and on 2 February 2021. The interviews were 56 min and 35 s, and 61 min and 55 s long, respectively. The interviews ended when respondents reached a point of saturation. Four main themes emerged from the data, which can be used to answer the research question: capacity development created a state of flux, infection prevention was a crucial component to COVID-19 crisis response work, the management of staff was an essential facilitation of crisis response work, and the respondents’ personal experiences. Table 1 shows a summary of these results, depicting the themes under each main theme.

### 3.1. State of Flux

The nurse managers spoke unanimously of how their PHC clinics had been in a state of flux for the duration of the crisis. Each respondent spoke of how their clinics functioned as large PHC providers within their communities before the crisis. They experienced changes to patient flow, nurses’ roles, and communication, as portrayed by Table 2, which includes themes and examples of original quotes.

Each respondent told of how, in setting up their COVID-19 hubs, they directed their regular patients to other municipal PHC clinics. Provision of non-essential PHC services declined and, as treatment queues grew, the nurse managers were increasingly concerned for their local communities. One respondent said that it was unclear to her whether other PHC clinics served their regular patients in numbers that would reflect successful continuation of care. Telemedicine services helped ease the burden of the crisis on PHC provision and created opportunities for nurses to work remotely. Overloaded telephone services, however, highlighted the importance of equitable need-for-care assessments.

In one COVID-19 hub, nurses were responsible for work in supportive roles while physicians saw all patients alone. Another hub operated utilizing physician and nurse pair work, in which physicians saw all patients but had nurses documenting and assisting in the exam room. The last COVID-19 hubs operated somewhat normally, in that patients classified as mild cases were seen independently by nurses, with a low physician consultation threshold. According to most respondents, communication between different stakeholders had improved drastically, though capacity development and IPC initially paused certain organizational structures such as non-essential meetings and trainings. Only one respondent told of how she had her staff trained to treat ARI and COVID-19 patients during initial capacity development. The respondent who had provided this training to her nurses also had her nurses working independently.

### 3.2. Infection Prevention

Conversation surrounding IPC was extensive and included the following themes (depicted in Table 3): stakeholders involved in IPC, planning of IPC measures, protocols in IPC, and equipment used in IPC. Nurse managers identified both experts and staff as stakeholders in IPC. They identified the patient’s role in IPC indirectly when they identified triage nurses as crucial stakeholders because they guided patients on IPC. The experts involved were the municipalities’ epidemiological departments and infection control nurses, with whom the respondents collaborated. The nurse managers followed and communicated new IPC guidelines that their staff were then responsible for receiving and implementing. Strict sick leave protocols were crucial to mitigating exposure among staff, though syndromic surveillance was not mentioned. Three respondents firmly declared that they had not had nosocomial infections, while one respondent did not confirm or deny either way. As an example of the creative IPC solutions that nurse managers implemented, each had organized lunch breaks differently. One respondent implemented staggered lunch breaks but did not elaborate further. To facilitate potential contact tracing, one respondent had everyone eating in the same groups of six and one had their staff reserve their lunch times on a spreadsheet every morning. The fourth strategy was to organize lunch so that staff used different break rooms according to their daily work assignments. This required turning meeting rooms into break rooms containing refrigerators, microwaves, and water dispensers.

Equipment for IPC included PPE, disposable utensils, and ionic air purifiers. One municipality provided lunch to its COVID-19 hub HCWs, along with disposable utensils as an IPC measure. Only one respondent mentioned utilizing ionic air purifiers, which were placed in each exam room. The topic of PPE, on the other hand, was brought up numerous times, as respondents noted they were worried for the safety of their staff when the availability and effectiveness of PPE were uncertain. Neither interview included a discussion about the specifics of how frontline HCWs used PPE. All nurse managers mentioned receiving PPE stock from the National Emergency Supply Agency. One respondent said the supplies that came from there were outdated and smelled of cellar. The municipalities lifted the COVID-19 hubs onto a list of critical operations, which meant that they received necessary supplies before other clinics did.

Each room was cleaned and items that had accumulated over the years, such as books, were removed. The strategic restocking of rooms, flow of people, time spent within the facility, clean/dirty sides, separate areas for donning/doffing PPE, and separate entrances for COVID-19 hub patients were carefully considered. There were notable differences in how respondents had organized their waiting areas and in how they handled confirmed COVID-19 cases. One respondent said that confirmed cases waited in an empty room, from which they were escorted to the exam room. Another said that their confirmed cases were all seen in one exam room but did not mention if they were waiting in a separate area. One respondent had separate waiting areas for patients over the age of 70 and for younger patients, which was notable because this interview took place before the equitable administration of vaccines began. Another respondent had arranged for patients to wait in separate areas according to their triage status.

### 3.3. Management of Staff

Nurse managers needed strong leadership skills and the ability to adjust their leadership styles, as they led, motivated, and recruited staff during an unprecedented global crisis. Table 4 shows examples of original quotes pertaining to the management of staff. In setting up the COVID-19 hubs, the very first thing respondents did was recruit the necessary nursing staff and address their concerns. Though all respondents considered working at the COVID-19 frontline as voluntary, some light contradictions to the voluntary nature of the work were evident. For instance, the authors use the term “recruit” lightly, as one respondent did not agree with the context of the term when verifying the results, though she too spoke of transferring at-risk staff to other PHC clinics. Need for further recruitment was at times exacerbated by low work motivation and the burnout of nursing staff. A shortage of nurses and workspaces affected the recruitment process as well.

At the point of the interviews, the respondents considered motivating staff to be their single biggest task and for this they relied heavily upon their own leadership skills, as one respondent stated clearly that she had not been provided with additional resources. The respondents stated that their nurses were motivated by occasional refreshments provided by municipalities, patients, and pharmaceutical companies. All respondents spoke highly of their staff and of how team spirit was high among frontline nurses. As managers, they were able to delegate responsibilities without hesitation, which also had a motivational impact. According to the majority of respondents, the most challenging time for staff motivation was after the summer of 2020 and into the fall, as their staff began losing hope. Both internal and external work rotations were the most utilized and most effective way to motivate staff.

Respondents told of how they had to adjust their leadership styles to better accommodate the complex needs of their staff, especially at the beginning of the crisis. One respondent told of how she had to strengthen her ability to recognize individual needs among her staff, as experienced nurses suddenly had more reservations than new nurses. Leading crisis response work at the COVID-19 hubs required skills such as mindful presence, openness, and listening, which all communicated availability. Clear communication skills quickly became critical, as respondents had to consistently communicate new information in a manner that was clear and easy to understand. By focusing on the clarity and transparency of communication, respondents were able to reassure nursing staff in times of uncertainty.

### 3.4. Personal Experiences

The general storyline to the “birth” of COVID-19 hubs states that the respondents had a few days’ time to first dismantle normal PHC operations and then set up COVID-19 hub operations. One original quote depicting this story is below in Table 5, under the theme of how COVID-19 hubs were born. One respondent heard of the decision to dedicate her clinic as a COVID-19 hub through the local newspaper. She lightly criticized her city’s timeliness in communication, though she ultimately had a few more days to prepare than her interview partner had. The urgency and stress involved in setting up operations over a short period was emphasized in each interview.

The respondents spoke openly about the big emotions and concerns they experienced, of which uncertainty was most prevalent. The feeling of uncertainty was present even during the interviews, as some respondents noted that they did not know what the vision for crisis recovery work was. The respondents and their staff also experienced fear of the virus, disbelief and hopelessness due to PPE shortages, and worry about the effectiveness of PPE. One respondent went so far as to say that she felt responsible for the health of her staff. On the flip side, crisis response work had also been rewarding. All respondents spoke of how, together with their nurses, they did not want to give up or give in. Two respondents noted they felt honored to have had the once-in-a-lifetime experience of leading the frontline of such a global crisis.

## 4. Discussion

The purpose of this study was to describe the early nurse management experiences from PHC clinics within Greater Helsinki dedicated to caring for all ambulatory patients with possible COVID-19 symptoms. The objectives were to make PHC crisis response contributions known and to provide an IAR of crisis response efforts. Together with the emerging literature, the results reflect that the COVID-19 crisis has immeasurably burdened PHC, the extent of which is yet unknown [14,44,45]. Primary health care is the frontline defense for COVID-19 response work at the community level, though the contribution is underrepresented in the literature [5,9,10,12,14,46,48].

Nurse managers of COVID-19 hubs described capacity development as a rapid and hectic process. They had a few days’ time to develop COVID-19 hub services for their communities. This process led to a state of flux that has lasted over a year and continues to stress the provision of non-essential, yet necessary, PHC services [45]. While the one respondent’s comment regarding the number of patients waiting in treatment queues is subjective experience, it is clear that a vision for crisis recovery work is needed as nurses begin to face the tremendous treatment queues with limited healthcare services to offer [43]. The COVID-19 crisis can be our best opportunity to develop our healthcare systems to be more resilient and to better address healthcare inequities overall [10,14,44,46,47].

Infection prevention was a major topic within both interviews, as the purpose of the COVID-19 hubs was to prevent infection. The implementation of IPC required creativity from the nurse managers as they planned and orchestrated different methods for the practical implementation of IPC measures within their own COVID-19 hubs. As the COVID-19 crisis has increased our understanding of the holistic consequences of uncontrolled pandemics, it has also created a fantastic opportunity for increasing IPC awareness globally [58]. Awareness comes at a critical point in global health, as the WHO has identified its first-ever list of priority pathogens, for which it recommends increasing and standardizing IPC practices, along with better coordination of surveillance [59]. Discrepancies within IPC, as seen in the literature and in this study, suggest that our efforts to standardize IPC practices must begin within our local communities [21,23,29,30].

The nurse managers of the four COVID-19 hubs spoke openly about their experiences during the interviews, often modeling servant leadership characteristics [41,42]. They have been on the frontline of the crisis and appreciate the unique opportunity it has afforded them. The respondents’ experiences, together with the literature, highlight the importance of leadership skills within COVID-19 response work [39,40,42]. Respondents spoke of how their single most important job has been and continues to be in motivating staff in the continued crisis response and recovery work, which is an experience supported by the literature as well [40]. As leadership in nursing faces new challenges brought forth by the crisis, nurse managers need more support in motivating nursing staff in order to mitigate the so-called COVID-19 effect, which may result in nurses leaving the profession [34,35,36,37,40].

### Limitations

Limitations to this study are related to the nature of the prolonged COVID-19 crisis and the possible language barrier. As a non-native Finnish speaker, the first author held both interviews in Finnish between 8 and 10 months into the crisis, with a 10-week gap between the two interviews due to the time required to secure research permits and schedule interviews. Neither the possible language barrier nor the contextual difference between the two interviews were seen to be of significance in the collected data. While piloting the interview is essential in thematic interviewing, this interview was not piloted due to the narrow target group, nature of the crisis, and the context of IARs [53]. Including respondents outside of the target group would have diluted the contextual base and this study would also have used more resources [60]. The results of this study are dependent upon the context. Trustworthiness for thematic analysis is determined through concepts of credibility, transferability, dependability, and confirmability. The engagement period was short, though data triangulation, respondent saturation, and member checking aided in achieving credibility [56].

## 5. Conclusions

This study is significant in that it is an IAR of PHC crisis response in Greater Helsinki from a nursing perspective. Nurse managers have stood in the face of fear and uncertainty and have led their staff through the beginning of an unprecedented global health crisis. Primary healthcare leaders can utilize the results of IARs and AARs to critically analyze and strengthen crisis preparedness in PHC for future and inevitable pandemics [14].

## Figures and Tables

**Table 1 ijerph-19-04885-t001:** Themes and main themes.

Themes	Main Themes
Changes to patient flow Changes to nurses’ roles Changes to communication between stakeholders	State of flux created by capacity development
Stakeholders involved in IPC Planning of IPC Protocols for IPC Equipment for IPC	IPC as a crucial component of COVID-19 crisis work
Recruiting staff to COVID-19 work Motivation of staff Leadership skills needed	Management of staff as an essential facilitation of crisis work
How COVID-19 hubs were born Personal experiences	Respondents’ personal experiences

**Table 2 ijerph-19-04885-t002:** Examples of original quotes depicting state of flux.

Example of Original Quote	Themes
… we had been a big urgent care clinic… all patients needing urgent care came straight through the door and to the nurses…we used to have eight frontline nurses on Mondays receiving urgent care patients so the mass of patients was pretty big… informing the community… how could we reach at least some of them so that everybody doesn’t rush in [on Monday]… … for sure the biggest burden…is how do we get the burden from the non-essential care handled when we at some point get to that work… there is a large number of non-essential appointments waiting clients are waiting impatiently… and what kind of bombs will we get there… the thousands, no thousands is no longer sufficient, but tens of thousands of clients in the queue…	Changes to patient flow
… if there is a [mild case] then they go straight to the nurse, so essentially the same as during normal urgent care operations… a large portion of patients do not necessarily see a physician at all… the nurse assesses the need for care, of course with a low physician consultation threshold…	Changes to nurses’ roles
… normally meetings were around [the city] and now [I] don’t need to drive every day in different directions as everything happens on [telecommunication channel]…	Changes to communication between stakeholders

**Table 3 ijerph-19-04885-t003:** Examples of original quotes depicting IPC.

Example of Original Quote	Themes
… when a patient comes through the door… we have a nurse receiving them… takes hand sanitizer and patient puts on a mask and then hand sanitizer again and then they are directed to sit…	Stakeholders involved in IPC
… changed the placement of the chairs and tables and removed extra chairs… so that gathering restrictions are implemented…	Planning of IPC
… we have three [break rooms] at the moment and it is very clearly divided that when you are working here, then you go there to eat and when you are working over there, you eat over here…	Protocols for IPC
… we received from the security-of-supply centers these so-called supplies which were stored there at some point for this purpose but it became apparent that they too were slightly outdated…and masks smelled a bit like cellar…	Equipment for IPC

**Table 4 ijerph-19-04885-t004:** Example quotes depicting motivation of staff.

Example of Original Quote	Themes
… at our clinic everyone is [there] purely voluntarily… they have been asked and they have volunteered…	Recruiting staff to COVID-19 work
… not in any monetary way have we been able to motivate because… there is no promise of extra vacation or anything like that… it comes from within the staff and from [when] we try to support them to… trust themselves and others…	Motivation of staff
… I noticed that a leadership style in which I speak to the nurses as one team and where I treat nurses somehow equally… I had to change it… I end up individually taking and considering each nurse kind of like according to their personality… it somehow feels like I have developed sensory antenna which are still extremely sensitive to…who is burning out…	Leadership skills needed

**Table 5 ijerph-19-04885-t005:** Example quotes depicting respondents’ personal experiences.

Example of Original Quote	Theme
… we got the information on Friday afternoon the 13th of the third [month] that on Monday operations had to be completely different and we had until then to cancel or move all of Monday’s appointments from this facility to elsewhere…[reiterates] there was the weekend’s time… and all spaces needed to be handled again, all rooms had to be organized… we had the time to warn staff that afternoon that Monday would be new operations and everyone was asked to come in early to work… it was a big job over the weekend… and also all the procedures like how do we… actually handle and what and where… and how do we organize the whole [city’s] operations anew…	How COVID-19 hubs were born
… I can say that it has been a unique experience… I would not have believed a year ago that we would be in this kind of a situation that the entire world has gotten this virus… pretty cool that I have been able to experience this on the frontline…	Personal experiences

## Data Availability

The data presented in this study are available on request from the corresponding author. The data are not publicly available due to the risk of compromised anonymity of respondents and their employers.

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
