# Peer review of "Early Nurse Management Experiences from Finnish COVID-19 Hubs: An In-Action Review"

_ijerph, 2022, doi:10.3390/ijerph19084885_

Round 1

Reviewer 1 Report

This manuscript, “Early nurse management experienced from Finnish primary care clinics functioning as COVID-19 ambulatory care facilities: an in-action review, will be of interest to nurse managers especially those who have worked through the various waves of the pandemic in all parts of the world. The strength of this manuscript is that it frames the leadership required at the meso-level of primary care clinics which were functioning as COVID-19 ambulatory care facilities.

There are several gaps in this manuscript which I will highlight below:

  1. Although one of the authors indicated that they were a native English speaker, I would encourage the authors to have the manuscript reviewed by an editor because some of the words are not in the right tense (i.e. past tense required) and the monies/percentages are written as we would see in manuscripts originating from the province of Quebec in Canada.
  2. The purpose of the study was to describe the early nurse management experiences from PHC clinics in the greater Helsinki area dedicated to caring for all ambulatory patients with possible COVID-19 symptoms. However, the questions were not stated; and, although most of the elements of qualitative rigour were evident, others were not e.g. was saturation reached? was member checking done?.
  3. In the Materials and Methods Section, the authors state that “After the study was published in December, 2021, the original data from the interviews (recording, transcriptions, original codes) were destroyed.” There are two things here: 1. Has this paper been published before and if so, then I wonder whether it should be considered for publication in this journal?; 2. Was the original manuscript in Finnish and if so this should be so stated in the Materials and Methods Section.; and 3. From an ethical stand-point, the fact that all aspects of the data (recording, transcriptions, original codes) were destroyed after the study was published is less than optimal. Are there no expectations from a local Research Ethics Board about how long and in what form the data is to be kept for? I have a responsibility to keep the data for a minimum of five years post-publication in case there are questions about the data or the data analysis. All of this to say, this aspect of the manuscript would not be seen as being adherent to the ethical principles for publication as outlined by the Canadian Institutes of Health Research, Natural Sciences and Engineering Research Council of Canada, and Social Sciences and Humanities Research Council, Tri-Council Policy Statement: Ethical Conduct for Research Involving Humans, December 2018.
  4. In the Materials and Methods Section, I was not able to identify whether or not this project received a Letter of Exemption or a Certificate of Approval from an appropriate Research Ethics Board. This is usually stated at the end of the Materials and Methods Section.
  5. There was very little linkage between the Literature Review and the Discussion, but it would have been an opportunity to weave servant leadership throughout the document as the need for leadership skills was one of themes evolving from the data. Although there may not have been any other current literature to compare the experiences of the nurse managers with, there is bound to be some literature that links management considerations as seen by physicians or other members of the health care team even though they are not the same or do the same tasks, it would be important to see what others have learned in this area as a result of the pandemic. Maybe nurses are not so different and maybe it is the balancing act that nurse managers have to do that makes those experiences different.
  6. What would be the Conclusion or the Summary that evolved from the data? The Conclusion is usually 2-3 sentences.

I think that this work is important for nurse managers and for those who hire/engage nurse managers, but the aforementioned gaps need to be considered and addressed before it could be considered for publication.

Author Response

Thank you for your comments and useful feedback. We appreciate the time you have taken to review our manuscript!

In general, and according to all feedback received, we have made substantial changes to the manuscript, including changing the title from “Early nurse management experiences from Finnish primary care clinics functioning as COVID-19 ambulatory care facilities: an in-action review” to “Early nurse management experiences from Finnish COVID-19 hubs: an in-action review”. The term COVID-19 ambulatory care site (ACS) is now referred to as COVID-19 hub to remain congruent with vocabulary used in literature (specifically from the UK). The acronym ACS may also be confused with alternate care site, which implies something different within the context of the COVID-19 health care crisis.

Overall, we have shortened the paper from 16 pages/9577 words to 13 pages/7661words.

  1. Thank you for pointing out the grammar and language issues. The manuscript has now been reviewed by an English language editor and necessary changes have been made. The style used for the monies and percentages was the style that is commonly used in Europe. We inquired from the journal as to their preference on the decimal style and have changed it to the American style. You can find the acknowledgement for the language review under acknowledgements.
  2. The research question is: What were the early experiences of nurse managers of primary health care clinics dedicated to caring for patients with possible COVID-19 symptoms? We have added that into the text for clarity. Saturation in respondent recruitment, or number of interviews conducted, was not needed because the entire target group participated in the study. The target group was defined as the nurse managers of the four Greater Helsinki COVID-19 hubs. The respondents reached saturation in the interviews and the first author reached theoretical saturation in reviewing of the themes during data analysis. These were mentioned in the manuscript, but we have elaborated some on them. Member-checking was done, and it was referred to by different terms such as collaboration and validation, and when mentioning the language barrier and timing of translation. We have added a clearer statement of this and have changed the terminology to member-checking throughout.
  3. Thank you for this comment. This study was conducted as the first author’s master’s thesis in Global Health and Crisis Management. We have added this detail into the purpose statement paragraph, found just before materials and methods. The original thesis work is published in the Finnish thesis database Theseus (https://urn.fi/URN:NBN:fi:amk-2021122190318). The original thesis publication is in English. The issues concerning language is seen in materials and methods as well as under the limitations of the study. “Data collection and analysis were in Finnish and the first author translated the results herself, as she considers herself fluent in the language [64]. … The translation from Finnish to English was delayed for as long as possible due to the ontological significance of allowing respondents to validate results in their own native language [64].”

This manuscript is original and has not been published or submitted for publishing anywhere else. The fact that the original data has been destroyed is understandably an ethical issue and the decision to destroy the data was not made lightly. In considering the limited usability of the data and the potential for the compromission of respondent anonymity, the decision to destroy the original data was made. This means the interview recordings and transcriptions, as well as the original codes, which were in paper form, including excerpts from the original interview transcripts. This is not in contradiction to data management guidelines in Finland (https://www.fsd.tuni.fi/en/services/data-management-guidelines/data-management-planning/). The digital mind maps from which the results were deduced/written have not been destroyed and are held by the first author. This has been clarified in the text.

  1. The Human Sciences Ethics Committee of the Helsinki Region Universities of Applied Sciences is responsible for providing a statement of ethical acceptability for non-medical research with human participants where participation:
    1. does not include informed consent
    2. an intervention to the physical integrity of participants is involved,
    3. participants are under the age of 15 and are participating without the informed consent of their guardians
    4. participants would be subjected to strong stimuli
    5. there is risk of mental harm to the participants or their loved ones
    6. if participation could pose a threat to the participant or their loved ones

If by these standards the study is exempt from requiring an ethical acceptability statement from the committee, the committee does not provide exemption statements. If needed, the committee will simply issue statements explaining the Finnish system. https://www.metropolia.fi/en/rdi/ethics-committee This study was first approved by the university and the second author, after which the three municipalities approved the study, and then the participants all signed informed consent forms. The study permits from the three municipalities and the signed informed consent letters are with the first author and can be presented as needed. The permits are in Finnish but can be translated as needed.

The following statement has been added into the manuscript:

This study was exempt from an ethical statement from the Human Sciences Ethics Committee of the Helsinki Region of Universities of Applied Sciences; however, the first author retains the study permits obtained from the municipalities and the informed consent letters signed by the respondents.

  1. Thank you for this constructive comment as well! We have made significant changes to both the literature review and discussion. For example, one original quote has been added to the theme “patient flow” to better tie the discussion around the topic into the results and literature review. When one is so immersed in one’s own work, one does not easily see the fragmentation that may occur, so this was very helpful.
  2. A conclusion has been added.

Reviewer 2 Report

1. The introduction offers a concise statement of the problem of PHC crisis, The relevant literature on the subject is discussed and cited, and the suggested approach is understandable.

2. It is indeed necessary to protect the anonymity of the respondents and their employers in the study, and the authors may then try to present unlabeled interview data through descriptive basic statistics.

Author Response

Thank you for taking the time to review our manuscript. In general, and according to other feedback received, we have made substantial changes to the manuscript including the title. We changed the title from “Early nurse management experiences from Finnish primary care clinics functioning as COVID-19 ambulatory care facilities: an in-action review” to “Early nurse management experiences from Finnish COVID-19 hubs: an in-action review”.  The term COVID-19 ambulatory care site (ACS) is now referred to as COVID-19 hub to remain congruent with vocabulary used in literature (specifically from the UK). The acronym ACS may also be confused with alternate care site, which implies something different within the context of the COVID-19 health care crisis.

Overall, we have shortened the paper from 16 pages/9577 words to 13 pages/7661words.

  1. Thank you for this comment. We have condensed the introduction and literature review to a more reader friendly length according to other feedback received. Hopefully this has only improved the manuscript from your perspective as well. The approach has been modified some as well, with additions concerning the ethical board review statement and other details.
  2. We appreciate the suggestion of using descriptive statistics in presenting unlabeled interview data and we considered it as an option here. Unfortunately, this may still compromise the anonymity of the respondents and their employers because of the limited number of respondents. The three municipalities organized their COVID-19 hubs in different ways and the experiences of the nurse managers reflect that. This is mostly seen and used in describing the way in which work was organized within the sites, for instance the role of nurses. If descriptive statistics were to be used to describe the experiences around the role of nurses, the municipality with two care sites would be singled out and easily identifiable. Which again leads to easier identification of the two other municipalities and ultimately, all respondents. Instead of using descriptive statistics to describe the data, we have changed the wording of the results, so that there would not be a need to refer to two respondents as one. We have also removed the sentence referring to this.

Reviewer 3 Report

Thank you for the opportunity to review this wonderful study. It has very strong idea framing and is very well developed. Understanding the roles of nurses and their experiences in dealing with COVID-19 will help in developing the health care system in general as well as in specific aspects. The weakness of this paper is not in the development of the idea or quality of the methodology or contribution; the major concern is about the presentation of the paper. The following are my recommendations for modification:

  • The title is very long; a shorter title will be more attractive to the readers
  • The introduction is very thorough as it provides details about the impacts of COVID-19 on the healthcare system, roles of PHC, and protocols taken from the Finnish government. However, the introduction is very long (eight paragraphs). It must be shortened to three to five paragraphs to make it easier to read and more interesting for the readers.
  • The purpose of the study comes after the introduction; I strongly recommend moving it to the end of the literature review.
  • The literature review is also very thorough, providing relevant references, but is also very long (10 paragraphs). I recommend the authors restrict it to five paragraphs or fewer, with focus on what previous studies have found. Include statements about pandemic crisis management, the roles of PHC in managing and dealing with COVID-19 cases, the roles of nurses in this crisis, and gap analysis, placing the purpose of the study at the end.
  • The methods of this study are of high quality, but an ethical approval number must be added.
  • One comment applicable to all sections is that, while this might work for a big project like a thesis, explaining everything in excessive detail is not suitable for publication in journals. Making the article shorter will be of high value to the readers.
  • The limitations section, like other sections, is very long. In the last paragraph, “The first author is a native English speaker, who is fluent in Finnish” seems like a strength rather than a limitation.

Author Response

Thank you for your kind remarks and authentic feedback! We appreciate the time you have put into reading and reviewing our manuscript.

We changed the title from “Early nurse management experiences from Finnish primary care clinics functioning as COVID-19 ambulatory care facilities: an in-action review” to “Early nurse management experiences from Finnish COVID-19 hubs: an in-action review”. The term COVID-19 ambulatory care site (ACS) is now referred to as COVID-19 hub to remain congruent with vocabulary used in literature (specifically from the UK). The acronym ACS may also be confused with alternate care site, which implies something different within the context of the COVID-19 health care crisis.

Overall, we have shortened the paper from 16 pages/9577 words to 13 pages/7661words.

  • We shortened the introduction from 9 paragraphs/1047 words to 4 paragraphs/563 words.
  • We shortened the literature review from 10 paragraphs/1158 words to 5 paragraphs/834 words
  • We slightly shortened the materials and methods from 12 paragraphs/1133 words to 10 paragraphs/1088 words.
  • We shortened the results from 22 paragraphs/2988 words to 12 paragraphs/2424 words.
  • We shortened the discussion from 9 paragraphs/1202 words to 5 paragraphs/656 words, which includes a shortened limitations section from 3 paragraphs/477 words to 1 paragraph/163 words.
  • There is an added Conclusions section of 1 paragraph/73words.

The purpose of the study has been moved to the end of the literature review. We have included a statement about the original thesis work as well.

The literature review has been revised according to your suggestions, and the discussion has been improved to link the literature review and results better.

The Human Sciences Ethics Committee of the Helsinki Region Universities of Applied Sciences is responsible for providing a statement of ethical acceptability for non-medical research with human participants where participation:

  1. does not include informed consent
  2. an intervention to the physical integrity of participants is involved,
  3. participants are under the age of 15 and are participating without the informed consent of their guardians
  4. participants would be subjected to strong stimuli
  5. there is risk of mental harm to the participants or their loved ones
  6. if participation could pose a threat to the participant or their loved ones

If by these standards the study is exempt from requiring an ethical acceptability statement from the committee, the committee does not provide exemption statements. If needed, the committee will simply issue statements explaining the Finnish system. https://www.metropolia.fi/en/rdi/ethics-committee This study was first approved by the university and the second author, after which the three municipalities approved the study, and then the participants all signed informed consent forms. The study permits from the three municipalities and the signed informed consent letters are with the first author and can be presented as needed. The permits are in Finnish but can be translated as needed.

The following statement has been added into the manuscript:

This study was exempt from an ethical statement from the Human Sciences Ethics Committee of the Helsinki Region of Universities of Applied Sciences; however, the first author retains the study permits obtained from the municipalities and the informed consent letters signed by the respondents.

The limitations section has been modified to include only limitations. It has also been shortened significantly.

Round 2

Reviewer 1 Report

I would like to congratulate the authors on making significant changes to the writing of the manuscript. The authors addressed many of my comments from the previous review; however I would encourage the authors to once again have an editor review the manuscript e.g. Line 86 - plexiglass partitions and delete of space; Lines 107 & 108 ... responding to the increased emotional health needs of nurses (not their nurses); Line 122 - seen in the literature. 

   The word epidemic is used several times in the Introduction but I think it should be replaced with pandemic as CoVID was not an epidemic as it impacted the entire country/world e.g. Line 47 - the first wave of the pandemic; Lines 50 & 51 - Countries that have been successful in slowing the progression of the CoVID-19 pandemic have implemented...   

Author Response

Thank you again for your comments! We really appreciate the time you dedicated to reading through the revised manuscript. We have had our editor look through the document once again and have made changes based on her suggestions. The choice to use “epidemic” instead of “pandemic” had been purposeful, in that Finnish authorities refer to the national pandemic situation as epidemic. As an example, the health authorities use the term in the link below. You are right though, as the targeted audience for this manuscript is international. We have changed the term “epidemic” to “pandemic”. One “epidemic” remains, but that is related to a study on nurses’ experiences from epidemics. Thank you for pointing that out to us!

https://thl.fi/en/web/infectious-diseases-and-vaccinations/what-s-new/coronavirus-covid-19-latest-updates/situation-update-on-coronavirus/the-covid-19-epidemic-regional-situation-recommendations-and-restrictions

Kind Regards,

Mari Nevala and Satu Vuorela